# A Joint Matrix Factorization Analysis of Multilingual Representations

**Zheng Zhao    Yftah Ziser    Bonnie Webber    Shay B. Cohen**

Institute for Language, Cognition and Computation

School of Informatics, University of Edinburgh

10 Crichton Street, Edinburgh, EH8 9AB

{zheng.zhao,yftah.ziser,bonnie.webber}@ed.ac.uk, scohen@inf.ed.ac.uk

## Abstract

We present an analysis tool based on joint matrix factorization for comparing latent representations of multilingual and monolingual models. An alternative to probing, this tool allows us to analyze multiple sets of representations in a joint manner. Using this tool, we study to what extent and how morphosyntactic features are reflected in the representations learned by multilingual pre-trained models. We conduct a large-scale empirical study of over 33 languages and 17 morphosyntactic categories. Our findings demonstrate variations in the encoding of morphosyntactic information across upper and lower layers, with category-specific differences influenced by language properties. Hierarchical clustering of the factorization outputs yields a tree structure that is related to phylogenetic trees manually crafted by linguists. Moreover, we find the factorization outputs exhibit strong associations with performance observed across different cross-lingual tasks. We release our code to facilitate future research.[1]

## 1 Introduction

Pre-trained multilingual models (Conneau and Lample, 2019a; Conneau et al., 2020; Liu et al., 2020; Xue et al., 2021) have gained widespread adoption in recent years. They initially pre-trained in many languages and subsequently fine-tuned for specific downstream tasks. Their aim is to leverage the linguistic knowledge acquired from similar languages, thereby benefiting low-resource languages and enabling zero-shot cross-lingual transfer ability. While numerous prior works have demonstrated these models have such abilities (Gerz et al., 2018; Ziser and Reichart, 2018; Aharoni et al., 2019; K et al., 2020; Muller et al., 2021; Fujinuma et al., 2022; Qiu et al., 2023), there are still open questions about the nature of the linguistic knowledge these models possess and the extent to which they

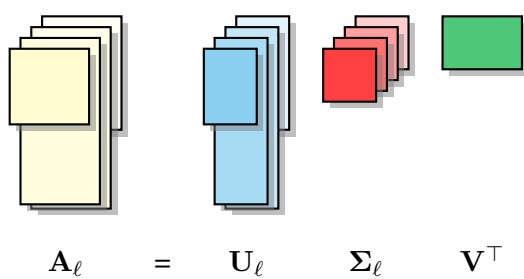

Figure 1: A diagram of the matrix factorization that PARAFAC2 performs. For our analysis, $\mathbf{A}_\ell$ ranges over covariance matrices between multilingual model representations and a $\ell$th monolingual model representations.

acquire and incorporate linguistic information in their multilingual representations.

Previous work has used singular vector canonical correlation analysis (SVCCA; Raghu et al. 2017) and other similarity statistics like centered kernel alignment (CKA; Kornblith et al., 2019) to analyze multilingual representations (Singh et al., 2019; Kudugunta et al., 2019; Muller et al., 2021). However, such methods can only compare one pair of representation sets at a time. In contrast to that, we analyze all multilingual representations simultaneously using parallel factor analysis 2 (PARAFAC2; Harshman 1972b), a method that allows us to factorize a set of representations jointly by decomposing it into multiple components that can be analyzed individually and then recombined to understand the underlying structure and patterns in the data. More precisely, we extend the sub-population analysis method recently presented by Zhao et al. (2022), who compare two models as an alternative to probing: a *control model* trained on data of interest and an *experimental model*, which is identical to the control model but is trained on additional data from different sources. By treating the multilingual experimental model as a shared component in multiple comparisons with different control models (each one is a monolingual model trained on

[1] https://github.com/zsquaredz/joint_multilingual_analysis/

a subset of the multilingual model), we can better analyze the multilingual representations.

As an alternative to probing, our representation analysis approach: a) enables standardized comparisons across languages within a multilingual model, circumventing the need for external performance upper bounds in making meaningful interpretation of performance metrics; b) directly analyzes model representations, avoiding the need for auxiliary probing models and potential biases from specific probing classifier architectures; and c) compares multilingual versus monolingual representations for any inputs, avoiding reliance on labelled probing datasets.

We use PARAFAC2 to directly compare representations learned between multilingual models and their monolingual counterparts. We apply this efficient paradigm to answer the following research questions on multilingual models: **Q1)** How do multilingual language models encode morphosyntactic features in their layers? **Q2)** Are our findings robust to address low-resource settings? **Q3)** Do morphosyntactic typology and downstream task performance reflect in the factorization outputs?

We experiment with two kinds of models, XLM-R (Conneau et al., 2020) and RoBERTa (Liu et al., 2019). We apply our analysis tool on the multilingual and monolingual representations from these models to check morphosyntactic information in 33 languages from Universal Dependencies treebanks (UD; Nivre et al., 2017a). Our analysis reinforces recent findings on multilingual representations, such as the presence of language-neutral subspaces in multilingual language models (Foroutan et al., 2022), and yields the following key insights:

- Encoding of morphosyntactic information is influenced by language-specific factors such as writing system and number of unique characters.
- Multilingual representations demonstrate distinct encoding patterns in subsets of languages with low language proximity.
- Representation of low-resource languages benefits from the presence of related languages.
- Our factorization method's utility reflects in hierarchical clustering within phylogenetic trees and prediction of cross-lingual task performance.

## 2 Background and Motivation

In this paper, we propose to use PARAFAC2 for multilingual analysis. By jointly decomposing a set of matrices representing cross-covariance between multilingual and monolingual representations, PARAFAC2 allows us to compare the representations across languages and their relationship to a multilingual model. For an integer $n$, we use $[n]$ to denote the set $\{1, ..., n\}$. For a square matrix $\mathbf{\Sigma}$, we denote by $\mathrm{diag}(\mathbf{\Sigma})$ its diagonal vector.

**PARAFAC2** Let $\ell$ index a set of matrices,[2] such that $\mathbf{A}_\ell = \mathbb{E}[\mathbf{X}_\ell \mathbf{Z}^\top]$, the matrix of cross-covariance between $\mathbf{X}_\ell$ and $\mathbf{Z}$, which are random vectors of dimensions $d$ and $d'$, respectively.[3] This means that $[\mathbf{A}_\ell]_{ij} = \mathrm{Cov}([\mathbf{X}_\ell]_i, \mathbf{Z}_j)$ for $i \in [d]$ and $j \in [d']$.

For any $\ell$ and two vectors, $\mathbf{a} \in \mathbb{R}^d, \mathbf{b} \in \mathbb{R}^{d'}$, the following holds due to the linearity of expectation: $\mathbf{a}\mathbf{A}_\ell \mathbf{b}^\top = \mathrm{Cov}(\mathbf{a}^\top \mathbf{X}_\ell, \mathbf{b}^\top \mathbf{Z})$. PARAFAC2 on the set of matrices $\{\mathbf{A}_\ell\}_\ell$ in this case finds a set of transformations $\{\mathbf{U}_\ell\}_\ell$, $\mathbf{V}$ and a set of diagonal matrices $\{\mathbf{\Sigma}_\ell\}_\ell$ such that:

$$\mathbf{A}_\ell \approx \mathbf{U}_\ell \mathbf{\Sigma}_\ell \mathbf{V}^\top. \tag{1}$$

We call the elements on the diagonal of $\mathbf{\Sigma}_\ell$ *pseudo-singular values*, in relationship to singular value decomposition that decomposes a single matrix in a similar manner. The decomposition in Eq. 1 jointly decomposes the matrices such that each $\mathbf{A}_\ell$ is decomposed into a sequence of three transformations: first transforming $\mathbf{Z}$ into a latent space ($\mathbf{V}$), scaling it ($\mathbf{\Sigma}_\ell$) and then transforming it into a specific $\ell$th-indexed $\mathbf{X}_\ell$ space ($\mathbf{U}_\ell$). Unlike singular value decomposition, which decomposes a matrix into a similar sequence of transformations with orthonormal matrices, PARAFAC2 does not guarantee $\mathbf{U}_\ell$ and $\mathbf{V}$ to be orthonormal and hence they do not represent an orthonormal basis transformation. However, Harshman (1972b) showed that a solution can still be found and is unique if we add the constraint that $\mathbf{U}_\ell^\top \mathbf{U}_\ell$ is constant for all $\ell$. In our use of PARAFAC2, we follow this variant. We provide an illustration of PARAFAC2 in Figure 1.

## 3 Experiment-Control Modeling for Multilingual Analysis

We employ factor analysis to generate a distinctive signature for a group of representations within an

---

[2]In our case, $\ell$ will usually index a language. We will have several collections of such matrices indexed by language, for example, slicing the representations from a specific morphosyntactic category. This becomes clear in §3 and §4.

[3]We assume that for any $\ell$, $\mathbb{E}[\mathbf{X}_\ell] = 0$ and $\mathbb{E}[\mathbf{Z}] = 0$.

experimental model, contrasting them with representations derived from a set of control models. In our case, the experimental model is a jointly-trained multilingual pre-trained language model, and the control models are monolingual models trained separately for a set of languages. Formally, there is an index set of languages $[L]$ and a set of models consisting of the experimental model $\mathbf{E}$ and the control models $\mathbf{C}_\ell$ for $\ell \in [L]$.

We assume a set of inputs we apply our analysis to, $\mathcal{X} = \bigcup_{\ell=1}^{L} \mathcal{X}_\ell$. Each set $\mathcal{X}_\ell = \{\mathbf{x}_{\ell,1}, \ldots, \mathbf{x}_{\ell,m}\}$ represents a set of inputs for the $\ell$th language. While we assume, for simplicity, that all sets have the same size $m$, it does not have to be the case. In our case, each $\mathcal{X}_\ell$ is a set of input words from language $\ell$, which results in a set of representations as follows. For each $\ell \in [L]$ and $i \in [m]$ we apply the model $\mathbf{E}$ and the model $\mathbf{C}_\ell$ to $\mathbf{x}_{\ell,i}$ to get two corresponding representations $\mathbf{y}_{\ell,i} \in \mathbb{R}^d$ and $\mathbf{z}_{\ell,i} \in \mathbb{R}^{d_\ell}$. Here, $d$ is the dimension of the multilingual model and $d_\ell$ is the dimension of the representation of the monolingual model for the $\ell$th language. Stacking up these sets of vectors separately into two matrices (per language $\ell$), we obtain the set of paired matrices $\mathbf{Y}_\ell \in \mathbb{R}^{m \times d}$ and $\mathbf{Z}_\ell \in \mathbb{R}^{m \times d_\ell}$. We further calculate the covariance matrix $\mathbf{\Omega}_\ell$, defined as: $\mathbf{\Omega}_\ell = \mathbf{Z}_\ell^\top \mathbf{Y}_\ell$.

**Use of PARAFAC2**  Given an integer $k$ smaller or equal to the dimensions of the covariance matrices, we apply PARAFAC2 on the set of joint matrices, decomposing each $\mathbf{\Omega}_\ell$ into:

$$\mathbf{\Omega}_\ell \approx \mathbf{U}_\ell \mathbf{\Sigma}_\ell \mathbf{V}^\top, \tag{2}$$

where $\mathbf{U} \in \mathbb{R}^{d_\ell \times k}$ and $\mathbf{V} \in \mathbb{R}^{d \times k}$.

To provide some intuition on this decomposition, consider Eq. 2 for a fixed $\ell$. If we were following SVD, such decomposition would give two projections that project the multilingual representations and the monolingual representations into a joint latent space (by applying $\mathbf{U}_\ell$ and $\mathbf{V}$ on $\mathbf{z}$s and $\mathbf{y}$s, respectively). When applying PARAFAC2 jointly on the set of $L$ matrices, we enforce the matrix $\mathbf{V}$ to be identical for all decompositions (rather than be separately defined if we were applying SVD on each matrix separately) and for $\mathbf{U}_\ell$ to vary for each language. We are now approximating the $\mathbf{\Omega}_\ell$ matrix, which by itself could be thought as transforming vectors from the multilingual space to the monolingual space (and vice versa) in three transformation steps: first into a latent space ($\mathbf{V}$), scaling it ($\mathbf{\Sigma}_\ell$), and then *specializing* it monolingually.

The diagonal of $\mathbf{\Sigma}_\ell$ can now be readily used to describe a *signature* of the $\ell$th language representations in relation to the multilingual model (see also Dubossarsky et al. 2020). This signature, which we mark by $\mathrm{sig}(\ell) = \mathrm{diag}(\mathbf{\Sigma}_\ell)$, can be used to compare the nature of representations between languages, and their commonalities in relationship to the multilingual model. In our case, this PARAFAC2 analysis is applied to different slices of the data. We collect tokens in different languages (both through a multilingual model and monolingual models) and then slice them by specific morphosyntactic category, each time applying PARAFAC2 on a subset of them.

For some of our analysis, we also use a condensed value derived from $\mathrm{sig}(\ell)$. We follow a similar averaging approach to that used by SVCCA (Raghu et al., 2017), a popular representation analysis tool, where they argue that the single condensed SVCCA score represents the average correlation across aligned directions and serves as a direct multidimensional analogue of Pearson correlation. In our case, each signature value within $\mathrm{sig}(\ell)$ from the PARAFAC2 algorithm corresponds to a direction, all of which are normalized in length, so the signature values reflect their relative strength. Thus, taking the average of $\mathrm{sig}(\ell)$ provides an intensity measure of the representation of a given language in the multilingual model. We provide additional discussion in §5.1.

## 4  Experimental Setup

**Data**  We use CoNLL's 2017 Wikipedia dump (Ginter et al., 2017) to train our models. Following Fujinuma et al. (2022), we downsample all Wikipedia datasets to an identical number of sequences to use the same amount of pre-training data for each language. In total, we experiment with 33 languages. For morphosyntactic features, we use treebanks from UD 2.1 (Nivre et al., 2017a). These treebanks contain sentences annotated with morphosyntactic information and are available for a wide range of languages. We obtain a representation for every word in the treebanks using our pre-trained models. We provide further details on our pre-training data and how we process morphosyntactic features in Appendix B.1.

**Task**  For pre-training our models, we use masked language modeling (MLM). Following Devlin et al. (2019), we mask 15% of the tokens. To fully control our experiments, we follow Zhao et al. (2022)

and train our models from scratch.

**Models** We have two kinds of models: the multilingual model $\mathbf{E}$, trained using all $L$ languages available, and the monolingual model $\mathbf{C}_\ell$ for $\ell \in [L]$ trained only using the $\ell$th language. We use the XLM-R (Conneau et al., 2020) architecture for the multilingual $\mathbf{E}$ model, and we use RoBERTa (Liu et al., 2019) for the monolingual $\mathbf{C}_\ell$ model. We use the base variant for both kinds of models. We use XLM-R's vocabulary and the SentencePiece (Kudo and Richardson, 2018) tokenizer for all our experiments provided by Conneau et al. (2020). This enables us to support all languages we analyze and ensure fair comparison for all configurations. We provide additional details about our models and training in Appendix B.2.

## 5 Experiments and Results

This section outlines our research questions (RQs), experimental design, and obtained results.

### 5.1 Morphosyntactic and Language Properties

Here, we address RQ1: *How do multilingual language models encode morphosyntactic features in their layers?* While in broad strokes, previous work (Hewitt and Manning, 2019; Jawahar et al., 2019; Tenney et al., 2019) showed that syntactic information tends to be captured in lower to middle layers within a network, we ask a more refined question here, and inspect whether different layers are specialized for specific morphosyntactic features, rather than providing an overall picture of all morphosyntax in a single layer. As mentioned in §3, we have a set of signatures, $\text{sig}(\ell)$ for $\ell \in [L]$, each describing the $\ell$th language representation for the corresponding morphosyntactic category we probe and the extent to which it utilizes information from each direction within the rows of $\mathbf{V}$. PARAFAC2 identifies a single transformation $\mathbf{V}$ that maps a multilingual representation into a latent space. Following that, the signature vector scales in specific directions based on their importance for the final monolingual representation it is transformed to. Therefore, the signature can be used to analyze whether similar directions in $\mathbf{V}$ are important for the transformation to the monolingual space. By using signatures of different layers in a joint factorization, we can identify comparable similarities for all languages. Analogous to the SVCCA similarity score (Raghu et al., 2017), we

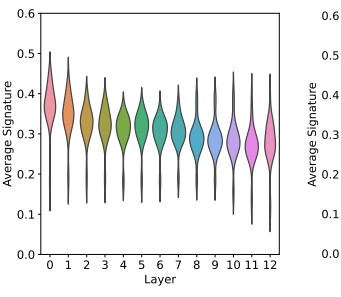
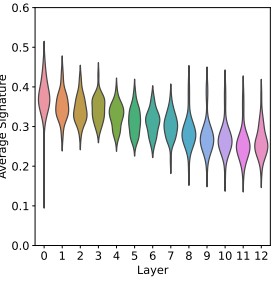

(a) Original data  (b) zh and ja romanized data

Figure 2: Average signature violin plots for all layers and languages on (a) original data and (b) data with Chinese (zh) and Japanese (ja) romanized.

condense each signature vector into a single value by taking the average of the signature. This value encapsulates the intensity of the use of directions in $\mathbf{V}$. A high average indicates the corresponding language is well-represented in the multilingual model. We expect these values to exhibit a general trend (either decreasing or increasing) going from lowers to upper layers as lower layers are more general and upper layers are known to be more task-specific (Rogers et al., 2020). In addition, the trend may be contrasting for different languages and morphosyntactic features.

**Language Signatures Across Layers** We begin by presenting the distribution of average $\text{sig}(\ell)$ values for all languages across all layers for all lexical tokens in Figure 2a. We observe a gradual decrease in the mean of the distribution as we transition from lower to upper layers. This finding is consistent with those from Singh et al. (2019), who found that the similarity between representations of different languages steadily decreases up to the final layer in a pre-trained mBERT model. We used the Mann-Kendall (MK) statistical test (Mann, 1945; Kendall, 1948) for individual languages across all layers. The MK test is a rank-based non-parametric method used to assess whether a set of data values is increasing or decreasing over time, with the null hypothesis being there is no clear trend. Since we perform multiple tests (33 tests in total for all languages), we also control the false discovery rate (FDR; at level $q = 0.05$) with corrections to the $p$-values (Benjamini and Hochberg, 1995). We found that all 33 languages except for Arabic, Indonesian, Japanese, Korean, and Swedish exhibit significant monotonically decreasing trends from lower layers to upper layers, with the FDR-adjusted $p$-values ($p < 0.05$). Figure 2a shows that the spread of the

distribution for each layer (measured in variance) is constantly decreasing up until layer 6. From these layers forward, the spread increases again. A small spread indicates that the average intensity of scaling from a multilingual representation to the monolingual representation is similar among all languages. This provides evidence of the multilingual model aligning languages into a *language-neutral* subspace in the middle layers, with the upper layers becoming more task-focused (Merchant et al., 2020). This result is also supported by findings of Muller et al. (2021) – different languages representations' similarity in mBERT constantly increases up to a mid-layer then decreases.

**Logogram vs. Phonogram**    In Figure 2a, we observe a long bottom tail in the average $\text{sig}(\ell)$ plots for all languages, with Chinese and Japanese showing lower values compared to other languages that are clustered together, suggesting that our models have learned distinct representations for those two languages. We investigated if this relates to the logographic writing systems of these languages, which rely on symbols to represent words or morphemes rather than phonetic elements. We conducted an ablation study where we romanized our Chinese and Japanese data into Pinyin and Romaji,[4] respectively, and retrained our models. One might ask why we did not normalize the other languages in our experiment to use the Latin alphabet. There are two reasons for this: 1) the multilingual model appears to learn them well, as evidenced by their similar signature values to other languages; 2) our primary focus is on investigating the impact of logographic writing systems, with Chinese and Japanese being the only languages employing logograms, while the others use phonograms. Figure 2b shows that, apart from the embedding layer, the average $\text{sig}(\ell)$ are more closely clustered together after the ablation. Our findings suggest that logographic writing systems may present unique challenges for multilingual models, warranting further research to understand their computational processes. Although not further explored here, writing systems should be considered when developing and analyzing multilingual models.

**Morphosyntactic Attributes**    Looking at individual morphosyntactic attributes, we observe that

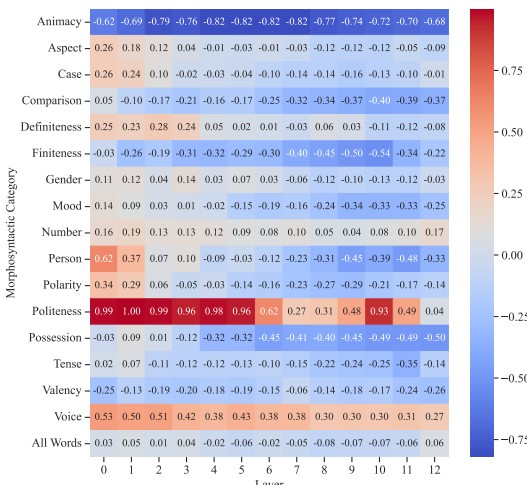

Figure 3: Pearson correlation results between the average $\text{sig}(\ell)$ for all languages and their data size for each morphosyntactic category among all layers.

while most attributes exhibit a similar decreasing trend from lower to upper layers, some attributes, such as Comparison and Polarity, show consistent distributions across all layers. Since these attributes occur rarely in our data ($< 1\%$ of tokens), it is possible that the model is only able to learn a general representation and not distinguish them among the layers. To investigate the effect of attribute frequency on our analysis, we performed a Pearson correlation analysis (for each attribute) between the average $\text{sig}(\ell)$ for all languages and their data size – the number of data points available in the UD annotations for a particular language and morphosyntactic feature. The results are shown in Figure 3. Our analysis of the overall dataset (all words) shows no evidence of correlation between attribute frequency and average $\text{sig}(\ell)$. However, upon examining individual categories, we observe a decrease in correlation as we move up the layers, indicating that the degree a morphosyntactic attribute is represented in the multilingual model is no longer associated with simple features like frequency but rather with some language-specific properties. This observation holds true for all categories, with the exception of Animacy, which is predominantly found in Slavic languages within our dataset. This aligns with the findings of Stanczak et al. (2022), who noted that the correlation analysis results can be affected by whether a category is typical for a specific genus. Next, we further explore the relationship between signature values and language properties.

**Language Properties**    In addition to data size, we explore the potential relationship between

---
[4]We use libraries available at: https://pypi.org/project/pypinyin/ and https://pypi.org/project/pykakasi/. We use the lazy_pinyin feature to generate Pinyins without tone marks.

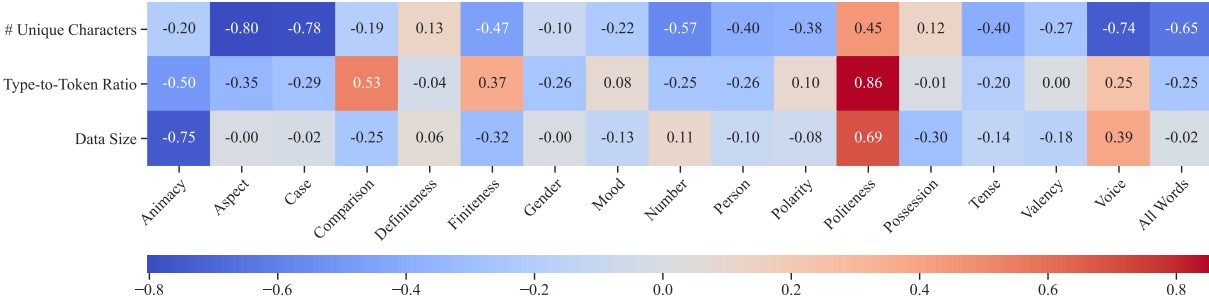

Figure 4: Pearson correlation results between the average $\text{sig}(\ell)$ for all languages and the number of unique characters, type-token ratio (TTR), and data size for each morphosyntactic category, averaged across all layers.

language-specific properties and the average $\text{sig}(\ell)$. We consider two language properties: the number of unique characters and the type-token ratio (TTR), a commonly used linguistic metric to assess a text's vocabulary diversity.[5] TTR is calculated by dividing the number of unique words (measured in lemmas) by the total number of words (measured in tokens) obtained from the UD annotation metadata. Typically, a higher TTR indicates a greater degree of lexical variation. We present the Pearson correlation, averaged across all layers, in Figure 4. To provide a comprehensive comparison, we include the results for data size as well. The detailed results for each layer can be found in Appendix C. Examining the overall dataset, we observe a strong negative correlation between the number of unique characters and signature values. Similarly, the TTR exhibits a similar negative correlation, indicating that higher language variation corresponds to lower signature values. When analyzing individual categories, we consistently find a negative correlation for both the number of unique characters and the TTR. This further supports our earlier finding that Chinese and Japanese have lower signature values compared to other languages, as they possess a higher number of unique characters and TTR.

**Generalization to Fully Pre-trained Models**  To ensure equal data representation for all languages in our experiment-controlled modeling, we down-sampled the Wikipedia dataset and used an equal amount for pre-training our multilingual models. To check whether our findings are also valid for multilingual pre-trained models trained on full-scale data, we conducted additional experiments

using a public XLM-R checkpoint.[6]  The setup remained the same, except that we used representations obtained from this public XLM-R instead of our own trained XLM-R. We observe that the trends for signature values were generally similar, except for the embedding and final layers, where the values were very low. This was expected, as the cross-covariance was calculated with our monolingual models. The similar trend among the middle layers further supports the idea that these layers learn language- and data-agnostic representations. Furthermore, the Pearson correlations between the number of unique characters, TTR, data size, and the average $\text{sig}(\ell)$ for the overall dataset were as follows: -0.65, -0.28, and -0.02, respectively. These values are nearly identical to those shown in Figure 4, confirming the robustness of our method and its data-agnostic nature.

## 5.2 Language Proximity and Low-resource Conditions

Here, we address RQ2: *Are our findings robust to address language subsets and low-resource settings?* In RQ1, our analysis was based on the full set of pre-training languages available for each morphosyntactic category we examine.  In this question, we aim to explore subsets of representations derived from either a related or diverse set of pre-training languages, and whether such choices yield any alterations to the findings established in RQ1.  Furthermore, we extend our analysis to low-resource settings and explore potential changes in results on low-resource languages, particularly when these languages could receive support from other languages within the same language family. We also explore the potential benefits of employing language sampling techniques for enhancing the representation of low-resource languages.

---

[5]To ensure accurate analysis, we filter out noise by counting the number of characters that account for 99.9% of occurrences in the training data. This eliminates characters that only appear very few times.

[6]https://huggingface.co/xlm-roberta-base

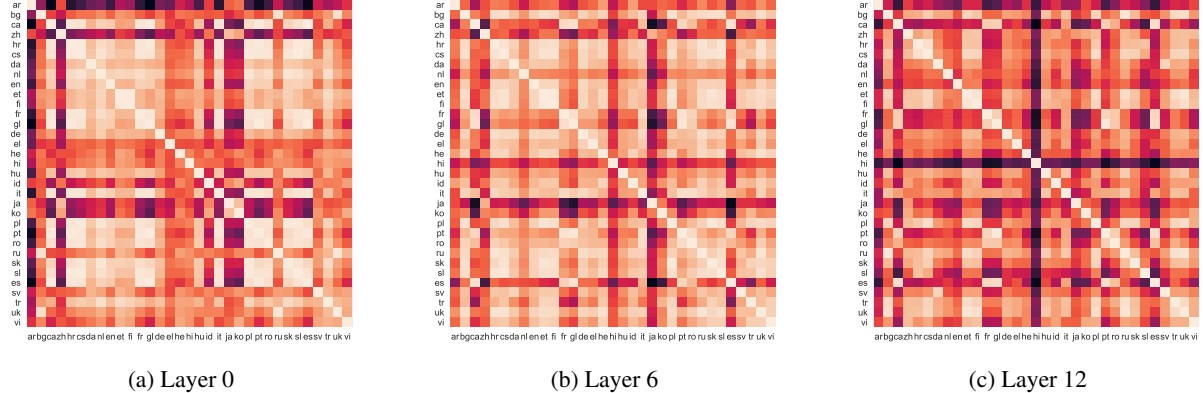

|          (a) Layer 0          |          (b) Layer 6          |          (c) Layer 12          |

Figure 5: Cosine distance matrices between all language pairs and their signature vectors based on overall representations obtained from layer 0, 6 and 12. Darker color indicates the cosine distance being close to 1.

**Language Proximity**  We obtain the related set of languages by adding all languages that are from the same linguistic family and genus (full information available in Appendix A). In total, we obtained three related sets of languages: Germanic languages, Romance languages, and Slavic languages. There are other related sets, but we do not include them in our experiment since the size of those sets is very small. For the diverse set of languages, we follow Fujinuma et al. (2022) and choose ten languages from different language genera that have a diverse set of scripts . These languages are Arabic, Chinese, English, Finnish, Greek, Hindi, Indonesian, Russian, Spanish, and Turkish. We use the $\chi^2$-square variance test to check whether the variance of the diverse set's average signatures from a particular layer is statistically significant from the variance of that of the related set, given a morphosyntactic category. We test layers 0 (the embedding layer), 6, and 12, covering the lower, middle, and upper layers within the model. We first find that for the overall dataset, the variance of the diverse set average signatures is significantly different (at $\alpha = 0.05$) from all three related set variances for all three layers. This suggests that, in general, multilingual representations are encoded differently for different subsets of languages with low language proximity. For the attributes of number, person, and tense, the variance within the diverse set significantly differs from the variances within the three related sets across all three layers, with a statistical significance level of $\alpha = 0.05$. This finding is sensible as all these three attributes have distinctions in the diverse set of languages. For example, Arabic has dual nouns to denote the special case of two persons, animals,

or things, and Russian has a special plural form of nouns if they occur after numerals. On the other hand, for attributes like gender, we do not witness a significant difference between the diverse set and related set since there are only four possible values (`masculine`, `feminine`, `neuter`, and `common`) in the UD annotation for gender. We speculate that this low number of values leads to low variation among languages, thus the non-significant difference. This finding concurs with Stanczak et al. (2022), who observed a negative correlation between the number of values per morphosyntactic category and the proportion of language pairs with significant neuron overlap. Hence, the lack of significant differences in variance between the diverse and related sets can be attributed to the substantial overlap of neurons across language pairs.

**Low-resource Scenario**  In order to simulate a low-resource scenario, we curtailed the training data for selected languages, reducing it to only 10% of its original size. The choice of low-resource languages included English, French, Korean, Turkish, and Vietnamese. English and French were selected due to the availability of other languages within the same language family, while the remaining languages were chosen for their absence of such familial relationships. Notably, Korean was specifically selected as it utilizes a distinct script known as Hangul. To examine the impact of low-resource conditions on each of the selected languages, we re-trained our multilingual model, with each individual language designated as low-resource. To address potential confounding factors, we also re-trained monolingual models on the reduced dataset. Additionally, we explored a sampling technique (Devlin, 2019) to enhance low-resource languages.

| Task | Dataset | #Lang. | Metric | mBERT | XLM | XLM-R | MMTE |
|------|---------|--------|--------|-------|-----|-------|------|
| Classification | XNLI (Conneau et al., 2018) | 12 | Acc. | .36 | .30 | .36 | .21 |
| | PAWS-X (Yang et al., 2019) | 7 | Acc. | .67 | .65 | .75 | .69 |
| Struct. pred. | POS (Nivre et al., 2017b) | 22 | F1 | .36 | .36 | .66 | .40 |
| | NER (Pan et al., 2017) | 22 | F1 | .46 | .46 | .55 | .46 |
| QA | XQuAD (Artetxe et al., 2020) | 10 | F1 / EM | .60 / .35 | .81 / .56 | .73 / .45 | .72 / .61 |
| | MLQA (Lewis et al., 2020) | 7 | F1 / EM | .23 / .31 | .46 / .48 | .64 / .68 | .28 / - |
| | TyDiQA-GoldP (Clark et al., 2020) | 6 | F1 / EM | .41 / .05 | .43 / .43 | .46 / .46 | .66 / .45 |
| Retrieval | BUCC (Zweigenbaum et al., 2017) | 4 | F1 | .72 | .96 | .83 | .63 |
| | Tatoeba (Artetxe and Schwenk, 2019) | 21 | Acc. | .15 | .24 | .28 | - |

Table 1: Pearson correlations between final layer's $\mathrm{sig}(\ell)$ and XTREME benchmark performances on various tasks.

Further details can be found in Appendix D.

Our analysis reveals the impact of low-resource conditions on signature values. English and French, benefiting from languages within the same language family, exhibit minimal changes in signature values, indicating a mitigation of the effects of low-resource conditions on language representation. Remaining languages without such support experience a significant decline in signature values (dropping from 0.3 to nearly 0), particularly after the embedding layer. This implies that low-resource languages struggle to maintain robust representations without assistance from related languages. Additionally, our findings suggest that language sampling techniques offer limited improvement in signature values of low-resource languages.

### 5.3 Utility of Our Method

Here we address RQ3: *Do morphosyntactic typology and downstream task performance reflect in the factorization outputs?* Having conducted quantitative analyses of our proposed analysis tool thus far, our focus now shifts to exploring the tool's ability to unveil morphosyntactic information within multilingual representations and establish a relationship between the factorization outputs and downstream task performance. To investigate these aspects, we conduct two additional experiments utilizing the signature vectors obtained from our analysis tool. Firstly, we construct a phylogenetic tree using cosine distance matrices of all signature vectors. Subsequently, we examine the correlations between the results of the XTREME benchmark (Hu et al., 2020) and the $\mathrm{sig}(\ell)$ values.

**Phylogenetic Tree** We first compute cosine distance matrices using all signature vectors for all 33 languages and 12 layers for each morphosyntactic attribute. Then, from the distance matrix, we use an agglomerative (bottom-up) hierarchical clustering method: unweighted pair group method with arith-

metic mean (UPGMA; Sokal and Michener, 1958) to construct a phylogenetic tree. We show the distance matrices between all language pairs and their signature vectors based on overall representations obtained from layers 0, 6 and 12 in Figure 5. We can observe that signatures for Arabic, Chinese, Hindi, Japanese, and Korean are always far with respect to those for other languages across layers. From the distance matrix, we construct a phylogenetic tree using the UPGMA cluster algorithm. We present our generated trees and a discussion in Appendix E.1. In short, the constructed phylogenetic tree resembles linguistically-crafted trees.

**Performance Prediction** To establish a robust connection between our factorization outputs and downstream task performances, we conducted an analysis using the XTREME benchmark, which includes several models: mBERT (Devlin et al., 2019), XLM (Conneau and Lample, 2019b), XLM-R, and MMTE (Arivazhagan et al., 2019). This benchmark encompasses nine tasks that span four different categories: classification, structured prediction, question answering, and retrieval. These tasks demand reasoning on multiple levels of meaning. To evaluate the relationship between the metrics of each task and our average $\mathrm{sig}(\ell)$ across all available languages for that task, we calculated the Pearson correlation. For each task's performance metrics, we use the results reported by Hu et al. (2020). The obtained correlation values using signature values from the last layer are presented in Table 1, along with pertinent details about each task, such as the number of available languages, and the metrics employed. For a comprehensive analysis, we also provide results using $\mathrm{sig}(\ell)$ from every layer in Appendix E.2. Observing the results, it becomes evident that the XLM-R model exhibits the highest correlation, which is expected since the $\mathrm{sig}(\ell)$ values obtained from our factorization process are also computed using the same architecture.

Furthermore, for most tasks, the highest correlation is observed with the final layers, which is reasonable considering their proximity to the output. Notably, we consistently observe high correlation across all layers for straightforward tasks like POS and PAWS-X operating on the representation level. However, for complex reasoning tasks like XNLI, only the final layer achieves reasonable correlation. These results suggest that the factorization outputs can serve as a valuable indicator of performance for downstream tasks, even without the need for fine-tuning or the availability of task-specific data.

## 6 Related Work

Understanding the information within NLP models' internal representations has drawn increasing attention in the community. Singh et al. (2019) applied canonical correlation analysis (CCA) on the internal representations of a pre-trained mBERT and revealed that the model partitions representations for each language rather than using a shared interlingual space. Kudugunta et al. (2019) used SVCCA to investigate massively multilingual Neural Machine Translation (NMT) representations and found that different language encoder representations group together based on linguistic similarity. Libovický et al. (2019) showed that mBERT representations could be split into a language-specific component and a language-neutral component by centering mBERT representations and using the centered representation on several probing tasks to evaluate the language neutrality of the representations. Similarly, Foroutan et al. (2022) employed the lottery ticket hypothesis to discover sub-networks within mBERT and found that mBERT is comprised of language-neutral and language-specific components, with the former having a greater impact on cross-lingual transfer performance. Muller et al. (2021) presented a novel layer ablation approach and demonstrated that mBERT could be viewed as the stacking of two sub-networks: a multilingual encoder followed by a task-specific language-agnostic predictor.

Probing (see Belinkov 2022 for a review) is a widely-used method for analyzing multilingual representations and quantifying the information encoded by training a parameterized model, but its effectiveness can be influenced by model parameters and evaluation metrics (Pimentel et al., 2020). Choenni and Shutova (2020) probed representations from multilingual sentence encoders and dis-

covered that typological properties are persistently encoded across layers in mBERT and XLM-R. Liang et al. (2021) demonstrated with probing that language-specific information is scattered across many dimensions, which can be projected into a linear subspace. Intrinsic probing, on the other hand, explores the internal structure of linguistic information within representations (Torroba Hennigen et al., 2020). Stanczak et al. (2022) conducted a large-scale empirical study over two multilingual pre-trained models, mBERT, and XLM-R, and investigated whether morphosyntactic information is encoded in the same subset of neurons in different languages. Their findings reveal that there is considerable cross-lingual overlap between neurons, but the magnitude varies among categories and is dependent on language proximity and pre-training data size. Other methods, such as matrix factorization techniques, are available for analyzing representations (Raghu et al., 2017; Morcos et al., 2018; Kornblith et al., 2019) and even modifying them through model editing (Olfat and Aswani, 2019; Shao et al., 2023a; Kleindessner et al., 2023; Shao et al., 2023b). When applied to multilingual analysis, these methods are limited to pairwise language comparisons, whereas our proposed method enables joint factorization of multiple representations, making it well-suited for multilingual analysis.

## 7 Conclusions

We introduce a representation analysis tool based on joint matrix factorization. We conduct a large-scale empirical study over 33 languages and 17 morphosyntactic categories and apply our tool to compare the latent representations learned by multilingual and monolingual models from the study. Our findings show variations in the encoding of morphosyntactic information across different layers of multilingual models. Language-specific differences contribute to these variations, influenced by factors such as writing systems and linguistic relatedness. Furthermore, the factorization outputs exhibit strong correlations with cross-lingual task performance and produce a phylogenetic tree structure resembling those constructed by linguists. These findings contribute to our understanding of language representation in multilingual models and have practical implications for improving performance in cross-lingual tasks. In future work, we would like to extend our analysis tool to examine representations learned by multimodal models.

## Limitations

Our research has several limitations. First, we only used RoBERTa and its multilingual variant, XLM-R, for our experiments. While these models are widely used in NLP research, there are other options available such as BERT, mBERT, T5, and mT5, which we have yet to explore due to a limited budget of computational resources. Second, to ensure equal data representation for all languages we experimented with, we downsampled Wikipedia resulting in a corpus of around 200MB per language. While we validated our findings against a publicly available XLM-R checkpoint trained on a much larger resource, further verification is still necessary. Third, our analyses are limited to morphosyntactic features, and in the future, we aim to expand our scope to include other linguistic aspects, such as semantics and pragmatics.

## Acknowledgments

This work was supported by the UKRI Centre for Doctoral Training (CDT) in Natural Language Processing through UKRI grant EP/S022481/1 and CDT funding from Huawei Technologies. We would like to thank Balint Gyevnar, Matthias Lindemann, Edoardo Ponti, Ivan Titov and the anonymous reviewers for their helpful feedback. We appreciate the use of computing resources through the CSD3 cluster at the University of Cambridge and the Baskerville cluster at the University of Birmingham.

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

# A    Information on Attributes and Languages

We first provide information about all languages we use in our experiment in Table 2. The information includes ISO 639-1 codes for all languages, the language family and the genus they belong to. In Table 3, we present all morphosyntactic attributes we experiment. For each attribute, we list all languages that have the attribute. We also provide a reverse list where we list by languages:

- **Arabic (ar)**: Aspect, Case, Definiteness, Finiteness, Gender, Mood, Number, Part of Speech, Person, Polarity, Politeness, Voice

- **Bulgarian (bg)**: Aspect, Case, Comparison, Definiteness, Gender, Mood, Number, Part of Speech, Person, Polarity, Tense, Valency, Voice

- **Catalan (ca)**: Aspect, Case, Definiteness, Finiteness, Gender, Mood, Number, Part of Speech, Person, Polarity, Possession, Tense

- **Chinese (zh)**: Aspect, Case, Number, Part of Speech, Person, Polarity, Valency, Voice

- **Croatian (hr)**: Animacy, Case, Comparison, Definiteness, Finiteness, Gender, Mood, Number, Part of Speech, Person, Polarity, Possession, Tense, Valency, Voice

- **Czech (cs)**: Animacy, Aspect, Case, Comparison, Finiteness, Gender, Mood, Number, Part of Speech, Person, Polarity, Possession, Tense, Valency, Voice

- **Danish (da)**: Case, Comparison, Definiteness, Finiteness, Gender, Mood, Number, Part of Speech, Person, Possession, Tense, Valency, Voice

- **Dutch (nl)**: Case, Comparison, Definiteness, Finiteness, Gender, Number, Part of Speech, Person, Tense, Valency

- **English (en)**: Case, Comparison, Definiteness, Finiteness, Gender, Mood, Number, Part of Speech, Person, Tense, Valency

- **Estonian (et)**: Aspect, Case, Comparison, Finiteness, Mood, Number, Part of Speech, Person, Polarity, Tense, Valency, Voice

- **Finnish (fi)**: Case, Comparison, Finiteness, Mood, Number, Part of Speech, Person, Polarity, Possession, Tense, Valency, Voice

- **French (fr)**: Aspect, Definiteness, Finiteness, Gender, Mood, Number, Part of Speech, Person, Polarity, Tense, Valency

- **Galician (gl)**: Part of Speech, Polarity

- **German (de)**: Case, Comparison, Definiteness, Finiteness, Mood, Number, Part of Speech, Person, Polarity, Politeness, Possession, Tense, Valency

- **Greek (el)**: Aspect, Case, Comparison, Definiteness, Finiteness, Gender, Mood, Number, Part of Speech, Person, Tense, Voice

- **Hebrew (he)**: Case, Definiteness, Finiteness, Number, Part of Speech, Person, Polarity, Possession, Tense, Valency, Voice

- **Hindi (hi)**: Aspect, Case, Finiteness, Gender, Mood, Number, Part of Speech, Person, Polarity, Politeness, Tense, Voice

- **Hungarian (hu)**: Case, Comparison, Definiteness, Finiteness, Mood, Number, Part of Speech, Person, Possession, Tense, Valency

- **Indonesian (id)**: Part of Speech, Polarity

- **Italian (it)**: Aspect, Comparison, Definiteness, Finiteness, Gender, Mood, Number, Part of Speech, Person, Tense

- **Japanese (ja)**: Part of Speech

- **Korean (ko)**: Part of Speech

- **Polish (pl)**: Animacy, Aspect, Case, Comparison, Finiteness, Gender, Mood, Number, Part of Speech, Person, Polarity, Possession, Tense, Valency, Voice

- **Portuguese (pt)**: Aspect, Case, Definiteness, Finiteness, Gender, Mood, Number, Part of Speech, Person, Polarity, Tense

- **Romanian (ro)**: Aspect, Case, Definiteness, Finiteness, Gender, Mood, Number, Part of Speech, Person, Polarity, Possession, Tense, Valency

- **Russian (ru)**: Animacy, Aspect, Case, Comparison, Finiteness, Gender, Mood, Number, Part of Speech, Person, Polarity, Tense, Valency, Voice

- **Slovak (sk)**: Animacy, Aspect, Case, Comparison, Finiteness, Gender, Mood, Number, Part of Speech, Person, Polarity, Possession, Tense, Valency, Voice

- **Slovenian (sl)**: Animacy, Aspect, Case, Comparison, Definiteness, Finiteness, Gender, Mood, Number, Part of Speech, Person, Polarity, Possession, Tense, Valency

- **Spanish (es)**: Aspect, Case, Comparison, Definiteness, Finiteness, Gender, Mood, Number, Part of Speech, Person, Polarity, Tense, Valency

- **Swedish (sv)**: Case, Comparison, Definiteness, Finiteness, Gender, Mood, Number, Part of Speech, Polarity, Tense, Voice

- **Turkish (tr)**: Aspect, Case, Mood, Number, Part of Speech, Person, Polarity, Politeness, Possession, Tense, Valency, Voice

- **Ukrainian (uk)**: Animacy, Aspect, Case, Comparison, Finiteness, Gender, Mood, Number, Part of Speech, Person, Polarity, Tense, Valency, Voice

- **Vietnamese (vi)**: Part of Speech, Polarity

Notice that in this list and in our work, we omit a language that has less than 100 instances labeled for a particular morphosyntactic category.

## B Additional Details for Experiments

### B.1 Details for Data

We use CoNLL's 2017 Wikipedia dump (Ginter et al., 2017) to train our models. Following Fujinuma et al. (2022), we downsample all Wikipedia datasets to an identical number of sequences in order to use the same amount of data for all language pre-training. The downsampled dataset is standardized to the Hindi corpus, which has the smallest size among all languages we examine. For each language's pre-training data, there are about 30M tokens (approximately 200MB). In total, we experiment with 33 languages. We provide the full list of languages used for our experiments in Appendix A. We also create a validation set with 1K

sequences (about 512 tokens per sequence) to measure model loss (cross-entropy) during pre-training. For morphosyntactic features, we use treebanks from UD 2.1 (Nivre et al., 2017a). These treebanks contain sentences annotated with morphosyntactic information and are available for a wide range of languages. We obtain a contextual representation for every word in the treebanks by feeding them to our multilingual/monolingual models. We then use the UniMorph schema (Kirov et al., 2018) to map each word with its parts of speech and morphosyntactic properties. We provide a list of morphosyntactic categories we use in Appendix A. We follow Stanczak et al. (2022) and use the converter (McCarthy et al., 2018) to switch morphosyntactic annotations from UD v2.1 to UniMorph schema.

### B.2 Details for Models

We use the XLM-R (Conneau et al., 2020) architecture for the multilingual $\mathbf{E}$ model, and we use RoBERTa (Liu et al., 2019) for the monolingual $\mathbf{C}_\ell$ model. We use the base variant for both kinds of models, which consists of 12 layers, 768 hidden dimensions, 8 attention heads for RoBERTa, and 12 attention heads for XLM-R. We use XLM-R's vocabulary and the SentencePiece (Kudo and Richardson, 2018) tokenizer for all our experiments provided by Conneau et al. (2020) in order to support all languages we analyze and enable fair comparison for all configurations. We do not use the original RoBERTa vocabulary and tokenizer since they only support English. We pre-train all models for a maximum of 150K steps, and all models use the validation set cross-entropy loss to perform early stopping. We use the AdamW optimizer (Loshchilov and Hutter, 2019) with a learning rate of $10^{-4}$. Our monolingual models were trained on four NVIDIA GeForce GTX 1080 Ti GPUs with a batch size of two per GPU, and our multilingual models were trained on four NVIDIA A100 GPUs with a batch size of 16 per GPU. Both models take about two days to train. We use PyTorch (Paszke et al., 2019), the HuggingFace library (Wolf et al., 2020) and the TensorLy library (Kossaifi et al., 2019) for all model implementation and PARAFAC2 computation.

## C Additional Results for RQ1

We present the Pearson correlations between the average $\text{sig}(\ell)$ for all languages and their number of unique characters for each morphosyntactic cat-

egory among all layers in Figure 6. The results for type-token ratio (TTR) is presented in Figure 7. Please note that in the figures, the last row is labeled as "All Words", representing the results obtained from using representations taken from the dataset with Part-of-Speech (PoS) attributes. Given that each lexical token in the dataset is associated with a PoS tag, this analysis encompasses the entire dataset, enabling us to observe and comprehend the overall trends captured in the representations.

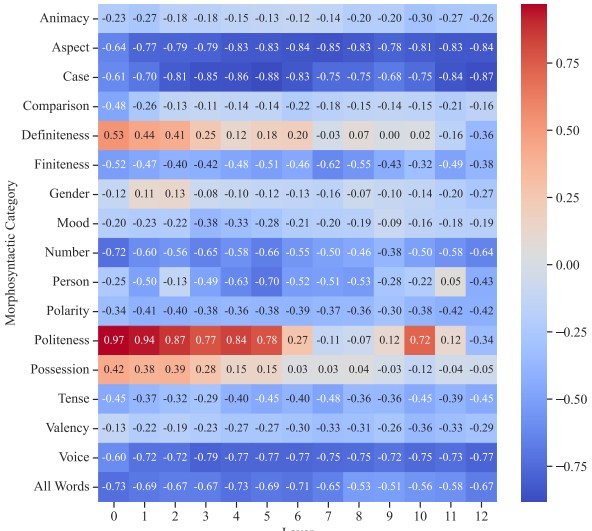

Figure 6: Pearson correlation results between the average $\text{sig}(\ell)$ for all languages and their number of unique characters for each morphosyntactic category among all layers.

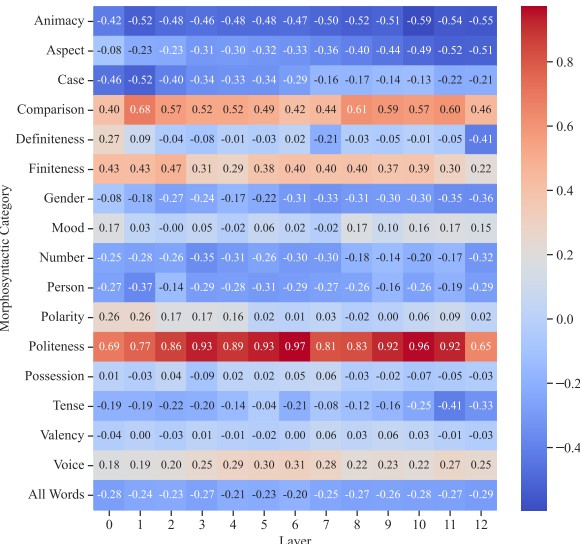

Figure 7: Pearson correlation results between the average $\text{sig}(\ell)$ for all languages and their type-token ratio (TTR) for each morphosyntactic category among all layers.

## D Additional Results for RQ2

Adhering to the current standard practice of language sampling during pre-training of multilingual models, we also experimented a setting inspired by the approach described by Devlin (2019). Following their approach, we applied a sampling technique to boost the representation of lower-resource languages. This involved sampling examples based on the probability $P(L) \propto |L|^\alpha$, where $P(L)$ represents the probability of selecting text from a given language during pre-training, and $|L|$ denotes the number of examples available in that language. For our study, we set the value of $\alpha$ to 0.3.

## E Additional Results for RQ3

We include in this section a phylogenetic tree analysis based on our approach, and a performance prediction experiment.

### E.1 Phylogenetic Tree

We show the tree generated from layer 6's matrix is provided in Figure 8a.

There is ongoing discussion over the specifics of the linguistic evolutionary phylogenetic tree of languages, and a tree model has limitations because not all evolutionary connections are fully hierarchical, and it is difficult to account for horizontal transmissions (Singh et al., 2019). Despite this, we can still see that the constructed phylogenetic tree closely matches the language tree that linguists created to describe the links and development of human languages. We can see that generally, Germanic, Romance, and Slavic languages are clustered in different sub-trees. In particular, West Slavic languages, South Slavic languages, and East Slavic languages are generally clustered together before being combined into the common Slavic language family. Also, Eastern Romance language Romanian are merged together with Western Romance languages to form the Romance language family cluster. Similar to the findings of Singh et al. (2019), we also observe that trees generated across different layers are generally similar. They may have different structures as the branching of the tree may differ, but languages within the same family or genus are also close in the tree.

So far, we have constructed trees based on the full slice of the data using representations from the PoS attribute. We also tried to generate trees using all other morphosyntactic attributes. However, since for most morphosyntactic attributes, some

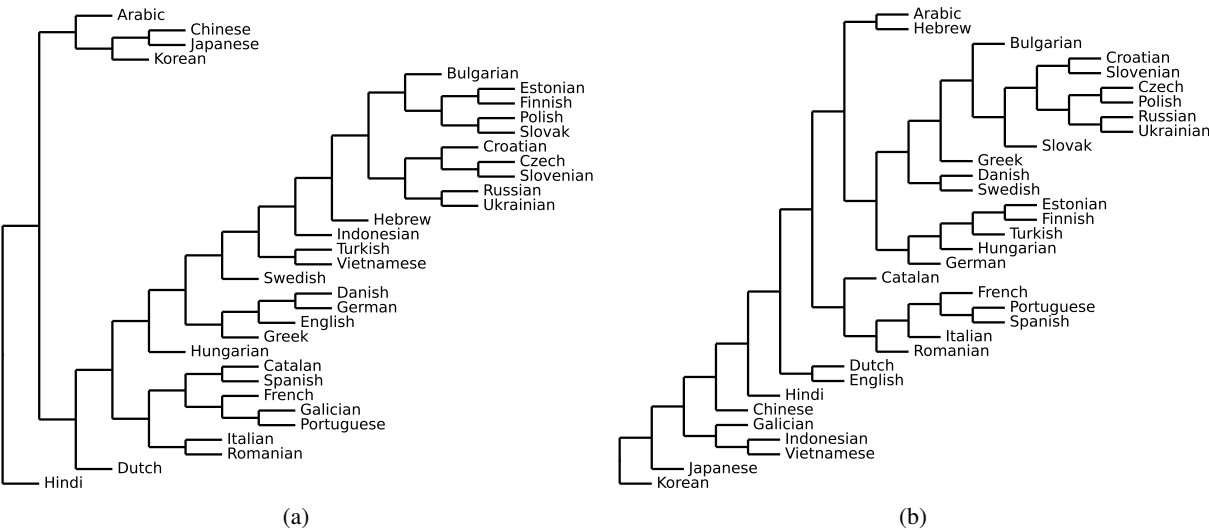

Figure 8: Phylogenetic trees of languages based on the distance between signatures vectors for all languages: (a) using overall representations; (b) averaged over morphosyntactic categories.

languages are always missing, i.e., the attribute is not available in that language, we construct an "average" tree by using the average distance matrices from all morphosyntactic categories except for PoS. If a language is missing in a category, we assign a distance of 1 to all other languages. We provide this average tree in Figure 8b. In this average tree, we observe a better fit, for example, Arabic and Hebrew are now in the same branch, and Chinese and Japanese are in their own branch as they belong to language families that are distinct from all other languages.

## E.2 Performance Prediction

We present correlations for all layers for mBERT, XLM, XLM-R, and MMTE in Table 4, Table 5, Table 6, and Table 7.

| Language | ISO 639-1 | Family | Genus |
|---|---|---|---|
| Arabic | ar | Afro-Asiatic | Semitic |
| Bulgarian | bg | Indo-European | Slavic |
| Catalan | ca | Indo-European | Romance |
| Chinese | zh | Sino-Tibetan | Chinese |
| Croatian | hr | Indo-European | Slavic |
| Czech | cs | Indo-European | Slavic |
| Danish | da | Indo-European | Germanic |
| Dutch | nl | Indo-European | Germanic |
| English | en | Indo-European | Germanic |
| Estonian | et | Uralic | Finnic |
| Finnish | fi | Uralic | Finnic |
| French | fr | Indo-European | Romance |
| Galician | gl | Indo-European | Romance |
| German | de | Indo-European | Germanic |
| Greek | el | Indo-European | Greek |
| Hebrew | he | Afro-Asiatic | Semitic |
| Hindi | hi | Indo-European | Indic |
| Hungarian | hu | Uralic | Ugric |
| Indonesian | id | Austronesian | Malayo-Sumbawan |
| Italian | it | Indo-European | Romance |
| Japanese | ja | Japanese | Japanese |
| Korean | ko | Korean | Korean |
| Polish | pl | Indo-European | Slavic |
| Portuguese | pt | Indo-European | Romance |
| Romanian | ro | Indo-European | Romance |
| Russian | ru | Indo-European | Slavic |
| Slovak | sk | Indo-European | Slavic |
| Slovenian | sl | Indo-European | Slavic |
| Spanish | es | Indo-European | Romance |
| Swedish | sv | Indo-European | Germanic |
| Turkish | tr | Altaic | Turkic |
| Ukrainian | uk | Indo-European | Slavic |
| Vietnamese | vi | Austro-Asiatic | Vietic |

Table 2: All languages used in this work along with language family and genus they belong to. Family and genus information is obtained from WALS.

| Property | Language |
|---|---|
| Animacy | bg, hr, cs, pl, ru, sk, sl, uk, |
| Aspect | ar, bg, ca, zh, hr, cs, et, fr, el, hi, hu, it, pl, pt, ro, ru, sk, sl, es, tr, uk |
| Case | ar, bg, ca, zh, hr, cs, da, nl, en, et, fi, fr, de, el, he, hi, hu, pl, pt, ro, ru, sk, sl, es, sv, tr, uk |
| Comparison | bg, hr, cs, da, nl, en, et, fi, fr, de, el, hu, it, pl, ro, ru, sk, sl, es, sv, uk |
| Definiteness | ar, bg, ca, hr, da, nl, en, fr, de, el, he, hu, it, pt, ro, sl, es, sv |
| Finiteness | ar, ca, hr, cs, da, nl, en, et, fi, fr, de, el, he, hi, hu, it, pl, pt, ro, ru, sk, sl, es, sv, uk |
| Gender | ar, bg, ca, hr, cs, da, nl, en, fr, el, hi, it, pl, pt, ro, ru, sk, sl, es, sv, uk |
| Mood | ar, bg, ca, hr, cs, da, en, et, fi, fr, de, el, he, hi, hu, it, pl, pt, ro, ru, sk, sl, es, sv, tr, uk |
| Number | ar, bg, ca, zh, hr, cs, da, nl, en, et, fi, fr, de, el, he, hi, hu, it, pl, pt, ro, ru, sk, sl, es, sv, tr, uk |
| Part of Speech | all 33 languages |
| Person | ar, bg, ca, zh, hr, cs, da, nl, en, et, fi, fr, de, el, he, hi, hu, it, pl, pt, ro, ru, sk, sl, es, tr, uk |
| Polarity | ar, bg, ca, zh, hr, cs, et, fi, fr, gl, de, he, hi, id, it, pl, pt, ro, ru, sk, sl, es, sv, tr, uk, vi |
| Politeness | ca, da, de, hi, es, tr |
| Possession | ar, ca, hr, cs, da, fi, de, he, hu, pl, ro, sk, sl, tr |
| Tense | bg, ca, hr, cs, da, nl, en, et, fi, fr, de, el, he, hi, hu, it, pl, pt, ro, ru, sk, sl, es, sv, tr, uk |
| Valency | bg, zh, hr, cs, da, nl, en, et, fi, fr, de, he, hu, pl, ro, ru, sk, sl, es, tr, uk |
| Voice | ar, bg, zh, hr, cs, da, et, fi, el, he, hi, pl, ru, sk, sv, tr, uk |

Table 3: All language properties (morphosyntactic categories) we analyzed in this work along with languages that have the corresponding property.

| Layer | Pair sentence | | Structured prediction | | Question answering | | | Sentence retrieval | |
|---|---|---|---|---|---|---|---|---|---|
| | XNLI | PAWS-X | POS | NER | XQuAD | MLQA | TyDiQA-GoldP | BUCC | Tatoeba |
| | Acc. | Acc. | F1 | F1 | F1 / EM | F1 / EM | F1 / EM | F1 | Acc. |
| 0 | 0.01 | 0.71 | 0.26 | 0.47 | 0.33 / 0.15 | 0.03 / 0.15 | 0.61 / 0.20 | 0.64 | -0.11 |
| 1 | -0.03 | 0.72 | 0.23 | 0.43 | 0.25 / -0.00 | -0.18 / -0.11 | 0.20 / -0.28 | 0.70 | 0.05 |
| 2 | 0.01 | 0.76 | 0.32 | 0.49 | 0.31 / 0.09 | -0.15 / -0.04 | 0.36 / -0.16 | 0.68 | 0.05 |
| 3 | 0.18 | 0.71 | 0.33 | 0.49 | 0.41 / 0.13 | -0.01 / 0.04 | 0.16 / -0.26 | 0.66 | 0.13 |
| 4 | 0.10 | 0.71 | 0.38 | 0.55 | 0.43 / 0.19 | -0.01 / 0.10 | 0.41 / -0.12 | 0.65 | 0.12 |
| 5 | 0.16 | 0.69 | 0.33 | 0.51 | 0.49 / 0.25 | 0.06 / 0.15 | 0.45 / -0.03 | 0.59 | 0.11 |
| 6 | 0.24 | 0.74 | 0.41 | 0.59 | 0.54 / 0.28 | 0.12 / 0.20 | 0.54 / 0.03 | 0.66 | 0.18 |
| 7 | 0.16 | 0.70 | 0.29 | 0.41 | 0.43 / 0.21 | 0.02 / 0.11 | 0.35 / -0.07 | 0.69 | 0.10 |
| 8 | 0.09 | 0.56 | 0.14 | 0.20 | 0.32 / 0.12 | -0.11 / -0.04 | 0.13 / -0.18 | 0.64 | 0.03 |
| 9 | 0.06 | 0.54 | 0.11 | 0.16 | 0.30 / 0.11 | -0.16 / -0.07 | 0.12 / -0.16 | 0.72 | -0.01 |
| 10 | 0.15 | 0.55 | 0.15 | 0.23 | 0.37 / 0.15 | -0.02 / 0.04 | 0.14 / -0.13 | 0.71 | 0.01 |
| 11 | 0.18 | 0.58 | 0.24 | 0.32 | 0.42 / 0.19 | 0.02 / 0.09 | 0.17 / -0.13 | 0.67 | 0.03 |
| 12 | 0.36 | 0.67 | 0.36 | 0.46 | 0.60 / 0.35 | 0.23 / 0.31 | 0.41 / 0.05 | 0.72 | 0.15 |

Table 4: Pearson correlations between $\mathrm{sig}(\ell)$ and XTREME benchmark performances of mBERT on various tasks.

| Layer | Pair sentence | | Structured prediction | | Question answering | | | Sentence retrieval | |
|---|---|---|---|---|---|---|---|---|---|
| | XNLI | PAWS-X | POS | NER | XQuAD | MLQA | TyDiQA-GoldP | BUCC | Tatoeba |
| | Acc. | Acc. | F1 | F1 | F1 / EM | F1 / EM | F1 / EM | F1 | Acc. |
| 0 | -0.01 | 0.68 | 0.28 | 0.53 | 0.61 / 0.38 | 0.24 / 0.29 | 0.66 / 0.67 | 0.90 | -0.02 |
| 1 | -0.11 | 0.71 | 0.25 | 0.52 | 0.51 / 0.22 | 0.03 / 0.06 | 0.59 / 0.53 | 0.96 | 0.13 |
| 2 | -0.07 | 0.74 | 0.34 | 0.57 | 0.54 / 0.27 | 0.05 / 0.11 | 0.65 / 0.60 | 0.97 | 0.12 |
| 3 | 0.09 | 0.70 | 0.37 | 0.55 | 0.65 / 0.36 | 0.23 / 0.22 | 0.42 / 0.36 | 0.98 | 0.21 |
| 4 | 0.05 | 0.70 | 0.39 | 0.59 | 0.68 / 0.40 | 0.21 / 0.27 | 0.58 / 0.54 | 0.94 | 0.20 |
| 5 | 0.09 | 0.67 | 0.34 | 0.54 | 0.72 / 0.46 | 0.27 / 0.31 | 0.50 / 0.48 | 0.93 | 0.18 |
| 6 | 0.17 | 0.72 | 0.41 | 0.63 | 0.75 / 0.48 | 0.32 / 0.35 | 0.67 / 0.65 | 0.94 | 0.27 |
| 7 | 0.08 | 0.68 | 0.29 | 0.46 | 0.69 / 0.45 | 0.26 / 0.28 | 0.53 / 0.51 | 0.95 | 0.19 |
| 8 | -0.02 | 0.54 | 0.16 | 0.24 | 0.55 / 0.34 | 0.12 / 0.13 | 0.29 / 0.26 | 0.97 | 0.09 |
| 9 | -0.04 | 0.51 | 0.12 | 0.20 | 0.53 / 0.32 | 0.09 / 0.11 | 0.25 / 0.23 | 0.92 | 0.06 |
| 10 | 0.07 | 0.52 | 0.17 | 0.26 | 0.61 / 0.38 | 0.23 / 0.22 | 0.29 / 0.28 | 0.92 | 0.08 |
| 11 | 0.10 | 0.56 | 0.25 | 0.34 | 0.65 / 0.40 | 0.25 / 0.26 | 0.33 / 0.31 | 0.97 | 0.11 |
| 12 | 0.30 | 0.65 | 0.36 | 0.46 | 0.81 / 0.56 | 0.46 / 0.48 | 0.43 / 0.43 | 0.96 | 0.24 |

Table 5: Pearson correlations between $\mathrm{sig}(\ell)$ and XTREME benchmark performances of XLM on various tasks.

| Layer | Pair sentence | | Structured prediction | | Question answering | | | Sentence retrieval | |
|---|---|---|---|---|---|---|---|---|---|
| | XNLI | PAWS-X | POS | NER | XQuAD | MLQA | TyDiQA-GoldP | BUCC | Tatoeba |
| | Acc. | Acc. | F1 | F1 | F1 / EM | F1 / EM | F1 / EM | F1 | Acc. |
| 0 | 0.11 | 0.80 | 0.67 | 0.62 | 0.63 / 0.34 | 0.59 / 0.62 | 0.58 / 0.52 | 0.91 | 0.14 |
| 1 | -0.03 | 0.80 | 0.61 | 0.57 | 0.51 / 0.14 | 0.39 / 0.40 | 0.61 / 0.47 | 0.84 | 0.21 |
| 2 | 0.02 | 0.83 | 0.67 | 0.64 | 0.58 / 0.24 | 0.46 / 0.50 | 0.63 / 0.47 | 0.85 | 0.23 |
| 3 | 0.13 | 0.78 | 0.66 | 0.62 | 0.55 / 0.19 | 0.43 / 0.45 | 0.37 / 0.16 | 0.85 | 0.19 |
| 4 | 0.15 | 0.80 | 0.73 | 0.68 | 0.71 / 0.39 | 0.56 / 0.61 | 0.59 / 0.46 | 0.90 | 0.31 |
| 5 | 0.17 | 0.78 | 0.69 | 0.63 | 0.70 / 0.40 | 0.61 / 0.63 | 0.51 / 0.43 | 0.93 | 0.28 |
| 6 | 0.24 | 0.82 | 0.73 | 0.70 | 0.73 / 0.42 | 0.63 / 0.64 | 0.68 / 0.62 | 0.89 | 0.35 |
| 7 | 0.15 | 0.78 | 0.62 | 0.53 | 0.57 / 0.26 | 0.52 / 0.55 | 0.53 / 0.46 | 0.87 | 0.25 |
| 8 | 0.04 | 0.66 | 0.47 | 0.33 | 0.36 / 0.05 | 0.32 / 0.37 | 0.30 / 0.23 | 0.88 | 0.13 |
| 9 | 0.03 | 0.63 | 0.44 | 0.30 | 0.34 / 0.03 | 0.29 / 0.35 | 0.26 / 0.19 | 0.85 | 0.10 |
| 10 | 0.13 | 0.64 | 0.49 | 0.36 | 0.41 / 0.10 | 0.37 / 0.41 | 0.29 / 0.24 | 0.86 | 0.11 |
| 11 | 0.19 | 0.67 | 0.56 | 0.43 | 0.52 / 0.20 | 0.45 / 0.49 | 0.36 / 0.34 | 0.85 | 0.15 |
| 12 | 0.36 | 0.75 | 0.66 | 0.55 | 0.73 / 0.45 | 0.64 / 0.68 | 0.46 / 0.46 | 0.83 | 0.28 |

Table 6: Pearson correlations between $\mathrm{sig}(\ell)$ and XTREME benchmark performances of XLM-R on various tasks.

| Layer | Pair sentence | | Structured prediction | | Question answering | | | Sentence retrieval | |
|---|---|---|---|---|---|---|---|---|---|
| | XNLI | PAWS-X | POS | NER | XQuAD | MLQA | TyDiQA-GoldP | BUCC | Tatoeba |
| | Acc. | Acc. | F1 | F1 | F1 / EM | F1 / EM | F1 / EM | F1 | Acc. |
| 0 | 0.02 | 0.73 | 0.33 | 0.51 | 0.51 / 0.53 | 0.06 / - | 0.92 / 0.35 | 0.59 | - |
| 1 | -0.10 | 0.75 | 0.26 | 0.46 | 0.43 / 0.37 | -0.15 / - | 0.82 / 0.00 | 0.61 | - |
| 2 | -0.06 | 0.78 | 0.38 | 0.53 | 0.48 / 0.44 | -0.14 / - | 0.93 / -0.04 | 0.58 | - |
| 3 | 0.02 | 0.73 | 0.36 | 0.51 | 0.54 / 0.42 | 0.07 / - | 0.79 / -0.32 | 0.55 | - |
| 4 | 0.01 | 0.74 | 0.44 | 0.58 | 0.59 / 0.52 | 0.00 / - | 0.91 / 0.06 | 0.58 | - |
| 5 | 0.08 | 0.71 | 0.39 | 0.54 | 0.64 / 0.57 | 0.07 / - | 0.85 / 0.22 | 0.53 | - |
| 6 | 0.15 | 0.76 | 0.44 | 0.61 | 0.69 / 0.59 | 0.13 / - | 0.91 / 0.31 | 0.58 | - |
| 7 | 0.07 | 0.73 | 0.35 | 0.43 | 0.59 / 0.57 | 0.09 / - | 0.78 / 0.26 | 0.61 | - |
| 8 | -0.02 | 0.59 | 0.21 | 0.23 | 0.46 / 0.44 | 0.00 / - | 0.59 / 0.14 | 0.55 | - |
| 9 | -0.04 | 0.56 | 0.19 | 0.21 | 0.44 / 0.44 | -0.04 / - | 0.54 / 0.16 | 0.66 | - |
| 10 | 0.05 | 0.58 | 0.20 | 0.27 | 0.51 / 0.46 | 0.10 / - | 0.55 / 0.20 | 0.64 | - |
| 11 | 0.08 | 0.61 | 0.31 | 0.34 | 0.55 / 0.49 | 0.11 / - | 0.54 / 0.28 | 0.57 | - |
| 12 | 0.21 | 0.69 | 0.40 | 0.46 | 0.72 / 0.61 | 0.28 / - | 0.66 / 0.45 | 0.63 | - |

Table 7: Pearson correlations between $\mathrm{sig}(\ell)$ and XTREME benchmark performances of MMTE on various tasks.