# OpenReview forum: "A Joint Matrix Factorization Analysis of Multilingual Representations"
_EMNLP/2023/Conference — EMNLP 2023 Findings_

### Official Review · Reviewer_ETeL · 2023-08-04

**Soundness:** 4

**Excitement:**

3: Ambivalent: It has merits (e.g., it reports state-of-the-art results, the idea is nice), but there are key weaknesses (e.g., it describes incremental work), and it can significantly benefit from another round of revision. However, I won't object to accepting it if my co-reviewers champion it.

**Missing References:**

[Choenni and Shutova, 2020](https://arxiv.org/abs/2009.12862)
 [Liang et al., 2021](https://arxiv.org/abs/2109.08040)

**Paper Topic And Main Contributions:**

This paper presents an analysis tool based on PARAFAC2 - a classic joint matrix factorization method for comparing latent representations of multilingual and monolingual language models -  to analyze the encoding of morphosyntactic features in representations. With this analysis tool, the authors try to answer the following research questions:
How do multilingual language models encode morphosyntactic features in layers  (Both in common pretraining settings and low-resource settings).
And what is the correlation between downstream task performance and the factorization outputs.

The authors pre-train multilingual and monolingual language models in XLM-R (RoBERTa) architecture with 33 languages from Wikipedia data. And conduct an empirical study of 17 morphosyntactic categories from UD treebanks by applying PARAFAC2 to analyze multilingual representations and monolingual representations.
They find variations in the encoding of morphosyntactic information across different layers, with category-specific differences influenced by language properties e.g. writing systems. The authors also show that the factorization outputs exhibit a strong correlation with performance observed across different cross-lingual tasks in the XTREME benchmark.

The main contributions of this paper include 1) proposing an analysis tool for multilingual representations; 2) the solid analysis and demonstration of morphosyntactic information in multilingual representations with language-specific differences; 3) the correlation analysis between factorization outputs and performance in cross-lingual tasks, which give an insight on practicality.

**Questions For The Authors:**

1) How to "enforce the matrix V to be identical for all decompositions"?

2) It's mentioned that you "downsample all Wikipedia datasets to an identical number of sequences in order to use the same amount of data". How do you make sure this is fair to all languages (if there is a reference setting)? Due to the difference in linguistic properties, some languages e.g. Chinese might have much fewer tokens. Would it be better to add another condition on size in MB?

3) How could you explain "We found that all 33 languages except for Arabic, Indonesian, Japanese, Korean, and Swedish exhibit significant monotonically decreasing trends from lower layers to upper layers"?

4) In your low-resource settings where the training data is "reducing it to only 10% of its original size", is there a reference to support this setting? In a reality setting e.g. XLMR is trained on some low-resource language with 0.1%-1% of the English data size, do you consider this kind of setting would have any impact on your findings or not?

5) How do you get performances on the XTREME benchmark? Directly copy from the XTREME paper or finetune the classifier by yourself?

6) It's stated that "These results suggest that the factorization outputs can serve as a valuable indicator of performance for downstream tasks, even without the need for finetuning or the availability of task-specific data." Could you provide a more general comparison - what are the computational requirements of these two approaches - to make this statement more convincing?

**Reasons To Accept:**

1) This paper proposes a different analytical tool based on joint matrix factorization, which enables a comprehensive comparison of multilingual representations and provides a unique perspective on the underlying structures and patterns.

2) The answers to the proposed research questions in this paper are logically clear. Analyzing how multilingual models encode morphosyntactic features and how they vary across languages (with different linguistic properties) and across different layers provides valuable insights. Especially, The unique analysis of logographic writing systems caused differences in Chinese and Japanese morphosyntactic features with those in other languages is inspiring and interesting.

3) The solid large-scale empirical research across multiple languages and morpho-syntactic categories enhances the generalizability of the findings.

4) The paper demonstrates the utility of analytical tools by showing the correlation between decomposition output and performance in cross-lingual tasks, which has practical implications for downstream task selection.

**Reasons To Reject:**

1) The paper is too dense and lacks clarity in explaining some technical details of the joint matrix factorization analysis e.g. How to "enforce the matrix V to be identical for all decompositions", and How to get token embedding (is it contextualized or the input is "<bos> token <eos>"). Please find more of these in the Questions.
More information (even in the appendix) on the specific implementation and algorithms used would be helpful for the readers to understand the analysis tool.

2) The paper does not provide a comprehensive quantitative discussion of the difference compared to other existing similar methods mentioned such as SVCCA (Raghu et al., 2017) and CKA (Kornblith et al., 2019), and not mentioned such as [Liang et al., 2021](https://arxiv.org/abs/2109.08040) and [Choenni and Shutova, 2020](https://arxiv.org/abs/2009.12862). Therefore, the advantages and disadvantages of the methods are not quantitatively compared.

3) The research scope of this paper is narrow and lacks novelty. The author did not quantitatively distinguish their proposed method from the previous related methods. The models used are limited (XLMR and mBERT), the latest encoder-decoder and decoder-only models are not involved. The adaptation to downstream tasks is limited to fine-tuning the classifier, did do not involve other novel finetuning and inference methods, such as cloze-style mask prediction, in-context learning, prompt-tuning, and prefix-tuning. It's an open question whether this method can be useful in a wider range of scenarios nowadays.

**Reproducibility:**

5: Could easily reproduce the results.

**Reviewer Confidence:**

4: Quite sure. I tried to check the important points carefully. It's unlikely, though conceivable, that I missed something that should affect my ratings.

---

> ### Author Rebuttal · Authors · 2023-08-28
>
> Thank you very much for your detailed review and for acknowledging the contributions presented in our work. We hope that the additional experiments we conducted have helped address your primary concerns regarding the lack of quantitative comparison with previous related methods.
>
> ### Reasons To Reject
>
> *__"The paper is too dense and lacks clarity in explaining some technical details of the joint matrix factorization analysis"__*
>
> In the revised version of the paper, we will enhance the algorithmic explanation by providing additional details in Sections 2 and 3. Moreover, we will add specifics regarding the PARAFAC2 implementation in the Appendix. As we aim to make our analysis tool accessible to the public, we will furnish comprehensive documentation, complete with illustrative examples, within the published repository. We note that PARAFAC2 is a well-known tensor factorization method, and we will include additional expository references.
>
> *__"The paper does not provide a comprehensive quantitative discussion of the difference compared to other existing similar methods mentioned such as SVCCA and CKA"__*
>
> We first would like to thank you for mentioning Liang et al., 2021 and Choenni and Shutova, 2020. Both of these works focus on analyzing multilingual representations with language-specific and their typological properties, which is relevant to our work. We will make sure we discuss them in our revised paper.
>
> However, it’s worth noting that both of these approaches rely on probing, a widely employed method for analyzing multilingual representations and quantifying the information encoded by training a parameterized model. As highlighted in our paper, the efficacy of probing can be influenced by model parameters and evaluation metrics [2, 3, 4, 5]. In contrast, our proposed methodology centers around matrix factorization, offering a distinct approach to representation analysis. While techniques like SVCCA and CKA share similarities, they are confined to pairwise comparisons, which may not be optimally suited for comprehensive analysis of multilingual representations.
>
> Taking your suggestion into consideration, we performed quantitative comparisons with our method by calculating correlations between SVCCA scores and XTREME benchmark scores (from XLM-R model). This analysis was conducted for XSQuAD, MLQA, and BUCC due to time limitations.
>
> | Method   | XSQuAD (F1) | MLQA (F1) | BUCC (F1) |
> |----------|-------------|-----------|-----------|
> | SVCCA    | 0.36        | 0.28      | 0.56      |
> | Ours     | **0.73**        | **0.64**      | **0.83**      |
>
> Encouragingly, our proposed method consistently demonstrates higher correlation scores in all cases when compared to SVCCA.
> We will add the results of the rest of the benchmarks upon acceptance.
>
> *__"The models used are limited (XLMR and mBERT), the latest encoder-decoder and decoder-only models are not involved. "__*
>
> While we acknowledge the importance of exploring various model architectures, as we have already highlighted in our limitations section, it's important to note that the primary emphasis of this work is not on exhaustive experimentation with all conceivable models but the novel multilingual analysis tool. Moreover, it's been observed that potent black-box LLMs like ChatGPT demonstrate shortcomings in comparison to more concise, task-specific models, especially when evaluating non-English scenarios [6]. Given that our study revolves around the analysis of multilingual capabilities and linguistic proficiency inherent in multilingual models, our focus remains on well-established models (e.g. XLM-R). An additional noteworthy point is the advantage of our analysis tool, which remains applicable as long as we can extract the desired linguistic features' representations from the model.
>
> *__"It's an open question whether this method can be useful in a wider range of scenarios nowadays, e.g., in-context learning, prompt-tuning, and prefix-tuning."__*
>
> In this work, we conducted tests of our multilingual analysis tool on models we trained from the ground up, as well as on publicly available checkpoints. These tests effectively underscore the resilience of our proposed analysis tool. As previously stated, the applicability of our tool extends as long as we can extract the intended representations from the model. In future work, we would like to investigate other inference methods such as prompting, in-context learning, and prefix-tuning.
>
> ### Questions For The Authors
> *__1: How to "enforce the matrix V to be identical for all decompositions"?__*
>
> The enforcement of V to be identical for all decompositions comes directly with the algorithm of PARAFAC2 [1]. We will make sure this is clearly explained in the revised paper.
>
> *__2: It's mentioned that you "downsample all Wikipedia datasets to an identical number of sequences in order to use the same amount of data". How do you make sure this is fair to all languages (if there is a reference setting)? Due to the difference in linguistic properties, some languages e.g. Chinese might have much fewer tokens. Would it be better to add another condition on size in MB?__*
>
> In our preprocessed Wikipedia dataset, each language’s data is processed by chunks with the same sequence length. We ensured that the downsampled dataset is similar in size and each language in the end has approximately 200 MB file size (mentioned in Appendix B.1). We will make sure we are clear about this in the revised version. Addressing your valid concern about variations in linguistic properties and the possibility of certain languages having fewer tokens or morphosyntactic features that could impact our factorization outcomes, we conducted an investigation into the potential influence of morphosyntactic feature frequency on our analysis (as detailed in Section 5.2). We did a Pearson correlation analysis between feature frequency and average signature values across all layers. Our findings yielded no conclusive evidence of a correlation between these factors, thereby assuaging concerns about potential bias arising from diverse feature frequencies across languages.
>
> *__3: How could you explain "We found that all 33 languages except for Arabic, Indonesian, Japanese, Korean, and Swedish exhibit significant monotonically decreasing trends from lower layers to upper layers"?__*
>
> While the observed trends do not attain statistical significance, it's noteworthy that some p-values approach the 0.05 threshold. We also postulate that these languages might lack significant representation of closely related siblings in the pretraining languages. This aligns with our prior experiment's observation that languages indeed benefit from related counterparts. Recognizing the intriguing nature of this observation, we intend to delve deeper into this aspect in our future research.
>
> *__4: In your low-resource settings where the training data is "reducing it to only 10% of its original size", is there a reference to support this setting? In a reality setting e.g. XLMR is trained on some low-resource language with 0.1%-1% of the English data size, do you consider this kind of setting would have any impact on your findings or not?__*
>
> We choose 10% since, in our low-resource experiment, we also pretrain the monolingual model from scratch using the 10% data for that language. Our initial trials using 1% or 0.1% data yielded an underfitting of the monolingual models. If our training capacity were expanded to 10 or 100 times the current scale, we would certainly incorporate these supplementary scenarios into our experimentation. However, the constraints of our computational resources currently prevent us from undertaking such extensive investigations. Additionally, we agree that in a realistic setup, languages are not represented equally in the data, and thus we experimented using a public checkpoint of XLM-R which we find that our findings are still robust (Section 5.1).
>
> *__5: How do you get performances on the XTREME benchmark?__*
>
> We get the results on the XTREME benchmark from their paper. We will be clear about this in our revised version.
>
> *__6: It's stated that "These results suggest that the factorization outputs can serve as a valuable indicator of performance for downstream tasks, even without the need for finetuning or the availability of task-specific data." Could you provide a more general comparison - what are the computational requirements of these two approaches - to make this statement more convincing?__*
>
> Our factorization outcomes carry significant potential for predicting performance, as they have showcased robust correlations with downstream task effectiveness (Section 5.3). To illustrate, consider a scenario involving a pre-trained multilingual model. Given knowledge of downstream performance for a set of languages, we aim to forecast the model's performance on language X—a language for which performance data is lacking. Traditionally, this process would demand task-specific data collection for language X, followed by GPU-based model fine-tuning and subsequent performance evaluation. However, with the aid of our analysis tool, this sequence simplifies remarkably. By extracting signature values through factorization (requiring only CPU resources), we can extrapolate language X's performance based on its signature values and the performances of other languages, streamlining the prediction process significantly.
>
> ### Missing References
> Finally, thank you for identifying the missing references, and we will make sure to include them in the revised paper.
>
> **References:**
>
> [1] Harshman 1972 https://psychology.uwo.ca/faculty/harshman/wpppfac2.pdf
>
> [2] Saphra and Lopez, 2019 https://aclanthology.org/N19-1329/
>
> [3] Zhang and Bowman, 2018 https://aclanthology.org/W18-5448/
>
> [4] Pimentel et al., 2020 https://aclanthology.org/2020.acl-main.420/
>
> [5] Zhao et al., 2022 https://aclanthology.org/2022.blackboxnlp-1.16/
>
> [6] Lai et al., 2023 https://arxiv.org/abs/2304.05613

---

### Official Review · Reviewer_3nrq · 2023-08-05

**Soundness:** 3

**Excitement:**

3: Ambivalent: It has merits (e.g., it reports state-of-the-art results, the idea is nice), but there are key weaknesses (e.g., it describes incremental work), and it can significantly benefit from another round of revision. However, I won't object to accepting it if my co-reviewers champion it.

**Paper Topic And Main Contributions:**

The paper applies a joint matrix factorization method, PARAFAC2, to analyzing multilingual representations in pertained models. The PARAFAC2 is similar to SVD, but the factorization is done jointly over all pairs of models across the multiple monolingual models and the multilingual model, with the constraint that the V matrix is shared across all factorizations. The paper then focuses on the language-specific \sigma matrices.

In particular, the paper shows that by analyzing the average signatures of the \sigma matrices, we can infer about the encoding of morphosyntactic information in the different layers of the pertained models. Additionally, the paper shows that the language-specific \sigma matrices can be used to interpret the similarities across languages, and the resulting phylogenetic tree appears to be linguistically viable. The factorization outputs can also be used to predict task performance across different languages.

**Questions For The Authors:**

1. How sensitive are the results to the choice of k (the "hyper parameter" in matrix factorization)?

2. How are the representations derived for each word? Are they feed into models one word at a time? Or do you take the contextual representations? It is not clear from the writing.

3. Is there any particular reason to use different model architecture for the monolingual models and the multilingual model (e.g., difference in number of attention heads).

**Reasons To Accept:**

- The application of joint matrix factorization, PARAFAC2, to the analysis of representations derived from a set of monolingual and a multilingual model is interesting. Previous matrix factorization techniques used in the literature for analyzing multilingual representations are typically restricted to analyzing one pair at a time. This works demonstrates a way to perform analysis across all language pairs at the same time.
- Some of the results from the analysis is interesting, for example, the phylogenetic tree derived from distance matrix, which is in turn constructed from the language-specific signature vectors, appears to be linguistically viable.

**Reasons To Reject:**

My major concern is the lack of clarity on the condensed values that are used across a number of analyses in the paper. The signature vectors (taken from the language-specific \sigma matrices) are condensed down to a single value by taking the average of the vector. And then the paper base a lot of analysis on top of these condensed values, e.g., look at the magnitude and spread of these values across languages and layers. It is not straightforward to understand what these average values mean (especially PARAFAC2 does not impose the same set of constraints as SVD). Maybe because it has been difficult to establish a clear idea of these condensed values, a lot of the following analyses and conclusions became hard to follow.  (Authors have responded below that this is following some prior practices to look at average correlations across aligned directions.)

**Reproducibility:**

3: Could reproduce the results with some difficulty. The settings of parameters are underspecified or subjectively determined; the training/evaluation data are not widely available.

**Reviewer Confidence:**

1: Not my area, or paper was hard for me to understand. My evaluation is just an educated guess.

**Typos Grammar Style And Presentation Improvements:**

It is not clear whether type-token ratio is computed based on word pieces or (UD-style) words.

---

> ### Author Rebuttal · Authors · 2023-08-28
>
> Thanks very much for your feedback and pointing out the key advantages of our proposed method compared with existing matrix factorization techniques.
>
> ### Reasons To Reject:
> *__"lack of clarity on the condensed values that are used across a number of analyses in the paper."__*
>
> We will make sure in the revised paper, we provide enough justification and explanation on taking averages on the signature values. We follow the same take average approach used by SVCCA [1], a very popular representation analysis tool, where they argue that the single condensed SVCCA score is the average correlation across aligned directions and is a direct multidimensional analogue of Pearson correlation. In our case, each pseudo singular value from our PARAFAC2 algorithm is associated with a direction, which are all normalized in length, and therefore, the pseudo singular values determine their strength. Thus, the average of the pseudo singular values gives the intensity of the representation of a specific language in the multilingual model.
>
> ### Questions For The Authors
>
> *__1: How sensitive are the results to the choice of k (the "hyper parameter" in matrix factorization)?__*
>
> The choice of k will certainly have some effect on the factorization process itself. In fact, setting k is not trivial for the PARAFAC2 algorithm [2]. That being said, based on our empirical observation, we noticed that a small portion of pseudo singular values of the components derived from the factorization tend to be exceedingly small, almost approaching 0. As a result, the impact of those singular values on our results (e.g. average signature values) is considered negligible.
>
> *__2: How are the representations derived for each word? Are they feed into models one word at a time? Or do you take the contextual representations?__*
>
> Following [3], we compute contextual representations for every individual word in the treebank by feeding them to our multilingual/monolingual models. We will make sure this is clear in the revised version of the paper.
>
> *__3: Is there any particular reason to use different model architecture for the monolingual models and the multilingual model (e.g., difference in number of attention heads).__*
>
> The standard base RoBERTa employs 8 attention heads, while XLM-R uses 12 heads. When training our XLM-R from scratch, we had the option to align the attention heads to 8, matching RoBERTa. However, since one of our experimental setups involves a public XLM-R checkpoint (with 12 heads), we chose 12 heads for comparability. Importantly, our method doesn't mandate that multilingual and monolingual models must share the same architecture, such as the number of hidden dimensions. The core principle of our approach is that each monolingual model is trained on a subset of data utilized for the multilingual model's training. Our method is versatile and can accommodate monolingual models with distinct architectures and hidden dimensions.
>
> ### Typos Grammar Style And Presentation Improvements
> *__"It is not clear whether type-token ratio is computed based on word pieces or (UD-style) words."__*
>
> The ratio is computed based on the token count provided in UD metadata (which is UD-style words). We will make sure this is clear in the revised paper.
>
> __References:__
>
> [1] Raghu et al., 2017 https://arxiv.org/abs/1706.05806
>
> [2] Bro, 1997 https://www.sciencedirect.com/science/article/abs/pii/S0169743997000324
>
> [3] Stanczak et al., 2022 https://aclanthology.org/2022.naacl-main.114/

---

### Official Review · Reviewer_85Fw · 2023-08-14

**Soundness:** 3

**Ethical Concerns:**

Yes

**Excitement:**

4: Strong: This paper deepens the understanding of some phenomenon or lowers the barriers to an existing research direction.

**Missing References:**

 the following references should be included in the bibliography:

- Devlin, J. (2019). BERT: Pre-training of deep bidirectional transformers for language understanding. arXiv preprint arXiv:1810.04805.
- Ginter, F., Hajiˇc, J., Luotolahti, J., Straka, M., & Zeman, D. (2017). CoNLL 2017 shared task - automatically annotated raw texts and word embeddings. LINDAT/CLARIAH-CZ digital library at the Institute of Formal and Applied Linguistics (ÚFAL), Faculty of Mathematics and Physics, Charles University.
- Sokal, R. R., & Michener, C. D. (1958). A statistical method for evaluating systematic relationships. University of Kansas science bulletin, 38(22), 1409-1438.
- Stanczak, K., Ponti, E., Henningen, L. T., Cotterell, R., & Augenstein, I. (2022). Same neurons, different languages: Probing morphosyntax in multilingual pre-trained models. In Proceedings of the 2022 Conference of the North American Chapter of the Association for Computational Linguistics: Human Language Technologies (pp. 1589-1598).
- Tenney, I., Das, D., & Pavlick, E. (2019). BERT rediscovers the classical NLP pipeline. In Proceedings of the 57th Annual Meeting of the Association for Computational Linguistics (pp. 4593-4601).

**Paper Topic And Main Contributions:**

This paper is about a Joint Matrix Factorization Analysis of Multilingual Representations. The authors present an analysis tool based on joint matrix factorization for comparing latent representations of multilingual and monolingual models. The paper addresses the problem of understanding how morphosyntactic features are reflected in the representations learned by multilingual pretrained models. The main contributions of this paper are the development of a novel analysis tool based on joint matrix factorization, a large-scale empirical study of over 33 languages and 17 morphosyntactic categories, and the demonstration of variations in the encoding of morphosyntactic information across different layers and influenced by language properties. The paper also provides insights into the relationship between morphosyntactic typology and downstream task performance, and the robustness of the findings in low-resource settings. The paper's contributions include computationally-aided linguistic analysis, NLP engineering experiments, and publicly available software and pre-trained models.

**Questions For The Authors:**

 Could you provide more details on the sampling technique explored in this paper to enhance low-resource languages? How does it work and what were the results of your experiments with it?

 How do you plan to extend the analysis tool based on joint matrix factorization developed in this paper to consider other linguistic features such as semantics or pragmatics?

 How do you envision the findings of this paper being applied in practical NLP applications, particularly for low-resource languages?

 Could you discuss any potential limitations or biases in the data used in this study, and how you addressed them in your analysis?

 How do you plan to make the software and pre-trained models developed in this paper publicly available, and what steps have you taken to ensure their usability and accessibility for researchers with varying levels of expertise?


**Reasons To Accept:**

The strengths of this paper include the development of a novel analysis tool based on joint matrix factorization, a large-scale empirical study of over 33 languages and 17 morphosyntactic categories, and the demonstration of variations in the encoding of morphosyntactic information across different layers and influenced by language properties. The paper also provides insights into the relationship between morphosyntactic typology and downstream task performance, and the robustness of the findings in low-resource settings.

If this paper were to be presented at the conference or accepted into Findings, the main benefits to the NLP community would be the availability of a new analysis tool for comparing latent representations of multilingual and monolingual models, a better understanding of how morphosyntactic features are reflected in the representations learned by multilingual pretrained models, and insights into the relationship between morphosyntactic typology and downstream task performance. The paper's contributions to computationally-aided linguistic analysis, NLP engineering experiments, and publicly available software and pre-trained models would also be valuable to the NLP community. Overall, this paper has the potential to advance the state-of-the-art in multilingual NLP and contribute to the development of more effective and efficient multilingual models.

**Reasons To Reject:**

One potential weakness of this paper is that the analysis is limited to morphosyntactic features and does not consider other linguistic features such as semantics or pragmatics. Additionally, the paper's focus on multilingual models may limit its generalizability to monolingual models or models trained on specific tasks.

The main risks of having this paper presented at the conference or accepted into Findings would be the potential for misinterpretation or overgeneralization of the findings. The paper's focus on multilingual models may lead some researchers to assume that the findings apply equally to monolingual models or models trained on specific tasks, which may not be the case. Additionally, the paper's emphasis on morphosyntactic features may lead some researchers to overlook other important linguistic features such as semantics or pragmatics. Finally, the paper's reliance on pre-trained models may limit its applicability to researchers who are developing their own models from scratch.

**Reproducibility:**

3: Could reproduce the results with some difficulty. The settings of parameters are underspecified or subjectively determined; the training/evaluation data are not widely available.

**Reviewer Confidence:**

3: Pretty sure, but there's a chance I missed something. Although I have a good feel for this area in general, I did not carefully check the paper's details, e.g., the math, experimental design, or novelty.

---

> ### Author Rebuttal · Authors · 2023-08-28
>
> Thank you very much for your feedback and recognition of our contributions.
>
> ### Reasons To Reject
> *__"The analysis is limited to morphosyntactic features and does not consider other linguistic features such as semantics or pragmatics."__*
>
> Over many years, linguists have worked with native speakers (both locally and via field work around the world) to come up with accurate characterizations of the world's languages. This work always starts by characterizing the morpho-syntactic properties of a language, as this is what is most evident in what they elicit from their native informants. Morphosyntactic attributes are important in NLP as they provide a deeper understanding of a language and what is encoded in words, including syntactic properties such as grammatical function and semantic properties such as agency, aspect, specificity, etc. It thus makes sense for an exploratory study of PARAFAC to use morphosyntactic features for three reasons: Such features are (i) semi-categorical, (ii) highly useful in many multilingual applications, such as grammatical error identification [1], PoS tagging [2], and dependency parsing [3], and (iii) available for more of the world's languages than any other properties.
>
> *__"The paper's focus on multilingual models may limit its generalizability to monolingual models or models trained on specific tasks"__*
>
> Our proposed analysis tool is to be used for analyzing multilingual models where an efficient comparison is being made between multilingual models and their monolingual counterparts. Our analysis tool is not limited to pre-trained models and can be easily applied to models trained on specific tasks.
>
> *__"The paper's reliance on pre-trained models may limit its applicability to researchers who are developing their own models from scratch"__*
>
> In our work, we actually train our models from scratch (Section 4), and we also show that our findings are robust with public pre-trained checkpoints (Section 5.1).
>
> ### Questions For The Authors
> *__Q: Could you provide more details on the sampling technique explored in this paper to enhance low-resource languages? How does it work and what were the results of your experiments with it?__*
>
> We will provide a more detailed account of sampling techniques in Appendix D. To briefly elaborate, when constructing the multilingual dataset, we sample each language’s data using a probability that is exponentially smoothed and proportional to their availability in the dataset. Thus low-resource languages will be over-sampled, and high-resource languages will be under-sampled. The results under low-resource settings are provided in Section 5.2. The signature values for English and French are very similar to the original setting, whereas for Korean, Turkish, and Vietnamese, the signature values dropped to nearly 0 after the embedding layer.
>
> *__Q: How do you plan to extend the analysis tool based on joint matrix factorization developed in this paper to consider other linguistic features such as semantics or pragmatics?__*
>
> Our analysis tool seamlessly accommodates the extension to other linguistic features. No modification of the analysis tool is necessary; as long as we possess the requisite data, we can readily input them into our analysis tool to derive corresponding factorization outcomes, such as signature values. As discussed above, one bottleneck of extending this work to other linguistic features, such as semantics and pragmatics, is the availability of data (especially availability for non-English languages). To start with, we can first extend to a smaller number of languages and analyze common semantic information like colour terms and kinship terms (which are commonly collected).
>
> *__Q: How do you envision the findings of this paper being applied in practical NLP applications, particularly for low-resource languages?__*
>
> The findings from our study in low-resource contexts propose a practical approach for enhancing low-resource languages within NLP applications. By incorporating linguistically related languages during pretraining, we can bridge data gaps and enrich the model's understanding of underrepresented languages. This strategy holds promise in various NLP tasks, like machine translation and sentiment analysis, by leveraging shared linguistic traits. Ultimately, this approach offers a pragmatic solution to empower low-resource languages and improve multilingual NLP outcomes. Furthermore, our study's insights can be harnessed to optimize the allocation of resources in NLP projects targeting low-resource languages. By identifying languages with linguistic affinities that offer the most substantial improvements in representation, researchers and practitioners can make informed decisions about where to focus efforts and resources, leading to more efficient and effective NLP solutions for languages with limited data.
>
> *__Q: Could you discuss any potential limitations or biases in the data used in this study, and how you addressed them in your analysis?__*
>
> As mentioned in the limitation, to ensure equal data representation for all languages we use in the experiment, we downsampled our data which resulted in about 200 MB per language. In reality, multilingual models will not be trained on an equal amount of data per language, so we have actually validated our findings using a public checkpoint of XLM-R which has been trained on a much larger resource with an unequal amount of data per language. We find that our findings are still robust with this model (Section 5.1). Furthermore, recognizing the inherent variance in linguistic properties, we scrutinized the potential influence of morphosyntactic feature frequency on our analysis (Section 5.2). Pearson correlation analysis revealed no significant correlation between feature frequency and average signature values across layers, addressing concerns of bias from varying feature frequencies across languages.
>
> *__Q: How do you plan to make the software and pre-trained models developed in this paper publicly available, and what steps have you taken to ensure their usability and accessibility for researchers with varying levels of expertise?__*
>
> We intend to host the software and pre-trained models on GitHub. This will allow researchers to easily access, download, and contribute to the project. We will provide extensive documentation accompanying the software and models. This documentation will encompass installation guides, usage instructions, code examples, and explanations of key functionalities. By offering clear and concise documentation, we aim to cater to both newcomers and experienced practitioners.
>
> ### Missing References
> Thank you for identifying these and we will incorporate the missing references in the revised version of the paper.
>
> __References:__
>
> [1] Pratapa el al., 2021 https://aclanthology.org/2021.emnlp-main.570/
>
> [2] Plank and Klerke, 2019 https://aclanthology.org/W19-6103/
>
> [3] Agic et al., 2014 https://aclanthology.org/W14-4203/

---

### Meta-Review · Area_Chair_QeqP · 2023-09-19

**Recommendation:** 4

**Metareview:**

The reviewers agree on the main reasons to accept:

* Novel analysis approach that compares across all language pairs at the same time.
* Large-scale empirical study.
* Insights into the relationship between morphosyntactic typology and downstream task performance.

However, they are mixed on excitement, due perhaps in part to a perceived lack of clarity in the paper's technical details.

---

### Decision · Program_Chairs · 2023-10-07

**Decision:**

Accept-Findings

**Comment:**

The reviewers agree on the main reasons to accept:

* Novel analysis approach that compares across all language pairs at the same time.
* Large-scale empirical study.
* Insights into the relationship between morphosyntactic typology and downstream task performance.

However, they are mixed on excitement, due perhaps in part to a perceived lack of clarity in the paper's technical details.